# Effects of Post-Harvest Elicitor Treatments with Ultrasound, UV- and Photosynthetic Active Radiation on Polyphenols, Glucosinolates and Antioxidant Activity in a Waste Fraction of White Cabbage (*Brassica oleracea* var. *capitata*)

**DOI:** 10.3390/molecules27165256

**Published:** 2022-08-17

**Authors:** Randi Seljåsen, Barbara Kusznierewicz, Agnieszka Bartoszek, Jørgen Mølmann, Ingunn M. Vågen

**Affiliations:** 1Department of Horticulture, Division of Food Production and Society, Norwegian Institute of Bioeconomy Research (NIBIO), P.O. Box 115, NO-1431 Ås, Norway; 2Department of Chemistry, Technology and Biotechnology of Food, Faculty of Chemistry, Gdansk University of Technology, 11/12 Narutowicza St., 80-233 Gdansk, Poland

**Keywords:** head cabbage, rest raw material, pre-processing, bioactive compounds, waste valorization

## Abstract

Biosynthesis of phytochemicals in leaves of *Brassica* can be initiated by abiotic factors. The aim of the study was to investigate elicitor treatments to add value to waste of cabbage. A leaf waste fraction from industrial trimming of head cabbage was exposed to UV radiation (250–400 nm, 59 and 99 kJ∙m^−2^, respectively), photosynthetic active radiation (PAR, 400–700 nm, 497 kJ∙m^−2^), and ultrasound in water bath (35 kHz, at 15, 30 and 61 kJ∙l^−1^ water), in order to improve nutraceutical concentration. UV was more effective than PAR to increase the level of flavonols (2 to 3-fold higher) and hydroxycinnamate monosaccharides (1 to 10-fold higher). PAR was three times as effective as UV to increase anthocyanins. Interaction of PAR + UV increased antioxidant activity (30%), the content of five phenolics (1.4 to 10-fold higher), and hydroxycinnamic monosaccharides (compared with PAR or UV alone). Indoles were reduced (40–52%) by UV, but the other glucosinolates (GLS) were unaffected. Ultrasound did not influence any parameters. The results are important for white cabbage by-products by demonstrating that UV + PAR can be successfully used as an effectual tool to increase important phenolics and antioxidant activity of waste fraction leaves without an adverse effect on the main GLS.

## 1. Introduction

In lieu of the growing population on the planet, wastage in food production represents a global food security problem and reduces revenues in the food sector. Annually, global food losses amount to about 1.3 billion tons, where root crops, fruits, and vegetables are among the most significant contributors with about 40–50% waste (http://www.fao.org/save-food/resources/keyfindings/en/ Access time: 1 July 2022). There is, therefore, a focus on reducing loss in food chains by utilizing waste raw materials for the development of novel food or feed products, pharmaceuticals, and other products [1]. The value of such waste fractions may be improved by various chemical or physical elicitor treatments, administered either pre- or post-harvest to enhance the contents of specific secondary metabolites [2,3].

Cabbage crops (*Brassica oleraceae* sp.) are among the most consumed vegetables worldwide. White cabbage is a widely used raw material in the food processing industry in salads or as juice and pre-cut ready-to-cook vegetables for consumer convenience and processed products. In the industrial processing of white cabbage, the raw material rest fraction may reach 30%, as seen, e.g., after trimming cabbage heads for industrial processing to the traditional product sauerkraut [4]. A similar trimming procedure is applied to stored cabbage before it is marketed for direct consumption. Such cabbage waste fractions, whether from the food industry or from trimming for fresh market, represent a raw material with the potential for increased value by enhancing its nutraceutical content.

Brassica vegetables’ health benefits include antioxidant, anticarcinogenic, and anti-inflammatory activities linked to the range of phenolic compounds, glucosinolates (GLS), and products of GLS degradation [5,6]. In the plant’s growth period, flavonoids’ concentration is most effectively increased by sunlight, while glucosinolates are more affected by temperature and sulphur supply [7]. Abiotic factors like high or low temperature, UV radiation, wounding, phytohormones, or altered gas composition in the environment can also be applied post-harvest to increase phenolics’ biosynthesis of anthocyanins and antioxidant activity [2,8].

In a recent study, Duarte-Sierra et al. [9] tested the evolution of glucosinolates and phenolic compounds (hydroxycinnamic acids) after UV-B radiation in stored broccoli at hormetic doses (1.5 kJ∙m^−2^) and higher doses (7.2 kJ∙m^−2^). The UV-B induced enhancement of hydroxycinnamic acids in their study was explained by a good connection between the expression of a specific gene (CYP79B3) and the level of indole glucosinolates, suggesting the target of UV-B to be a branch pathway of indole glucosinolates. In a recent study of stored broccoli [10], white light illumination (radiation spectra not specified) increased glucosinolate levels. At the same time, the light exposure showed lower expression of genes involved in glucosinolate biosynthesis. Thus, it was concluded that the glucosinolate degradation was lower than by storage in darkness. Post-harvest UV-B irradiation has also been shown to increase the content of flavonoids and phenolic acids in spinach, radish sprouts, parsley [11], and white cabbage [12]. 

Glucosinolates’ response to elicitor treatments is less straightforward than flavonoids as there are examples in the literature of plants where pre- or post-harvest exposure to UV-B radiation either increase [8,13], decrease [12,14], or does not influence [14] glucosinolate contents in intact Brassica plants. Glucosinolates are a group of thioglucosides containing a cyano- and a sulphate group and are grouped into mainly aliphatic and indolic glucosinolates based on amino-acid biosynthesis precursors [15]. 

Wave sonication (ultrasound) can, similarly to UV irradiation, act as abiotic stress for plants, according to the experimental evidence reviewed by da Silva and Dobránzki [16]. Ultrasound treatment had stimulating effects on producing secondary compounds like saponins in ginseng cells [17] and isoflavones in Genista tinctoria suspension cultures [18]. The sonication effects depend on the sound wave’s properties: frequency and intensity, pressure level, and duration of exposure [19]. Also, the plant species and variety (genotype) affect the sensibility of sonicated cells or tissues, as reviewed by Rokhina et al. [20]. So far, ultrasound, as an elicitor for secondary metabolites, has not been studied for Brassica. In studies on other plant species, leaves of lettuce exposed to ultrasound treatment (25kHz, APD of 26 W l-1 water) exhibited an increase in phenylalanine ammonia-lyase (PAL) activity after 60 h of storage, resulting in the production of phenolic compounds and enhancement of antioxidant activity [21].

The present study is an approach to reduce losses in the food chain by adding value to side streams (crop waste fractions), a topic of high interest for the food industry. Cabbage leaf rest fractions have not previously been studied with respect to phytochemical improvements by physical elicitor factors like ultrasound treatment (2–8 sec) in combination with UV irradiation (10–14 h) or photosynthetic active radiation (PAR). Ultrasound was selected as an appropriate method based on its ability to enrich nutraceuticals in lettuce leaves within a very short elicitation period (1–3 sec) [21]. PAR and UV were included as well-known phytochemical elicitors for plants. Still, there is a lack of studies on enhancement of phytochemical concentration to waste streams of pale inner leaves of stored cabbage (with restricted radiation exposure during cultivation). The compounds being analyzed in the present study, phenols, GLS, isothiocyanates, and indoles, are among the most studied Brassica phytonutrients that have the potential to be modified by the selected elicitors. For the economic justification in the food industry, it is important to find methods with the shortest possible time required for induction of the biochemical response, and to find methods that are possible to implement with industrial scale assembling bands or rotating drum systems where moving leaf parts are exposed continuously over time.

## 2. Materials and Methods

### 2.1. Plant Materials and Elicitor Treatment

Raw materials for the study were 30 kg leaves of head cabbage (*Brassica oleracea* var. *capitata*) c.v. ‘Bartolo F1’ (Bejo Seeds) originating from a co-stream of industrial processing of cabbage (Bama Industrier AS, Moss, Norway). The head cabbage raw material was cultivated by a local farmer on contract for Bama Industrier AS in the 2017 growing season (May to November), which had a mean temperature in the growing season of 13.5 °C (0.8 °C above average temperature for the last 30 years). The crop was harvested in December and stored in cold storage at 1 ± 1 °C on the farm until the 25th of February, where the products were transported by tractor-trailer to BAMA industry AS. The co-stream product from the processing of cabbage was detached leaves from the cleaning and sorting step of the industrial processing line. Leaves were sampled 30 min after cutting; packed in perforated PE sacks (15 kg each); transported by refrigerator car (4 ± 1 °C) to BAMA Storkjøkken AS, Lillesand, Norway; and thereafter picked up by car for a 15 min transport at ambient temperature to the experimental site at NIBIO’s (Norwegian Institute of Bioeconomy Research) research facility at Landvik, Grimstad, Norway. Prior to the start of the experiment, leaf fractions were stored in PE sacks in darkness at 4 °C for 4–6 days. The rest fraction consisted of whole leaves and fractured leaves. The elicitor experiments were performed in separate sections in a growth chamber at 18 ± 1 °C, at saturated humidity obtained by a lining of wet cloth on the table and a cover 4 cm above the leaves with cellulose acetate filter (CA-foil, crystal clear, 0.14 mm × 650 mm × 1200 mm: art nr. I81.1.1-1706 L, Nordbergstekniska AB, Vallentuna, Sweden). For each treatment, 20 randomly chosen pieces of leaves from the industry co-stream were exposed.

Elicitor treatments were carried out as described in Table 1, by combinations of ultrasound exposure (35 kHz, 15–61 kJ∙m^−1^ water), fluorescent UV radiation at low (59 kJ∙m^−2^) or high level (99 kJ∙m^−2^), and exposure to PAR (400–700 nm, 497 kJ∙m^−2^).

All treatments started by 2 h with either PAR (light-adapted leaves) or darkness (dark-adapted leaves). After that, ultrasound treatment (0, 2, 4 or 8 min) was performed, followed by 10 h exposure with UV and/or PAR, then concluded with another 2 h of PAR or darkness. After the treatments, all samples were left for 4–5 h in darkness before extracting samples for phytochemical analyses. The reference sample was leaves that received no elicitor treatments. Those leaves were kept in the dark in the experimental climate chamber during the testing period in a PE box (30 cm × 40 cm × 20 cm) covered with lightproof black PE foil (PE short day lining, LOG pr. no. 683172, Erik Storm Denmark). The experiment was repeated three times continuously over 3.5 days.

#### 2.1.1. Ultrasound Treatment

Since studies on ultrasound treatment of cabbage leaves are limited, exposure levels in the present study were based on effective doses given for lettuce [21], where phenolics and antioxidant activity were increased. Since lettuce has tender leaves, the dose was increased from 25 to 35 kHz due to the thicker leaves of cabbage. According to Xin et al. [22], a similar ultrasound dose (30 kHz) was effective in changing the microstructure of broccoli cells under freeze treatment in processing.

For each elicitor treatment, leaf fractions were exposed to ultrasound in an ultrasonic bath (DL510H DIGIPLUS-Sonorex, pr. no. 9877868, Heco Laboratorieutstyr AS) with stainless steel tank (9.7 litres, 30 cm × 24 cm × 15 cm). The ultrasonic bath frequency was 35 kHz, and the energy amount was 640 W (used at 100% level), given by four Lead Zirconate Titanate (PZT) broad beam transducers located at the bottom of the tank. Leaves were soaked in water by placing in one layer and cover with a metal grid located 2–3 cm below the water surface in a total volume of 5-litre water heated to 17 °C (±1 °C). Exposure durations were 0 (soaked in water only), 2, 4 or 8 min (15–61 kJ∙l^−1^ water), which was within the limit of no visible destructions of leaf cells. Leaves were treated in three portions of 6–7 leaf fractions in one layer in the ultrasound bath, which corresponded to 150–200 g at a time. After exposure, leaves were kept at 97–99% humidity and received treatments either including darkness or UV radiation, as described in sections below. Temperature increases of 1–2 °C were measured for the different treatment durations.

#### 2.1.2. Irradiation Treatment

Irradiation treatment was performed in a climate chamber in separate sections (65 cm × 120 cm × 80 cm), arranged in a line on a table, each with open parts on both short ends to facilitate the same temperature and humidity conditions in the radiation chamber sections. A number of 20 leaf parts covering an area of 30 cm × 115 cm were placed with the ventral or dorsal leaf side randomly facing upwards. The sections were lined with lightproof black PE foil (PE short day lining, LOG pr. no. 683172, Erik Storm Denmark). Fluorescent UV lamps (UV-B 313 EL, tube d 3.5 cm, length 120 cm, Elastocon AB, Brämhult, Sweden) were the source for UV exposure. Along with the UV lamps, PAR lamps (Philips TL-D 58W/33-640) were installed to provide photosynthetically active radiation. Lamps were placed in the chambers for the different treatments alternately, with the UV lamps in the middle and white lamps on both sides. For each chamber, UV lamps (0, 1 or 2 in one armature) and PAR lamps (0 or 2 in each of 2 armatures per treatment) were arranged to give the amount of radiation required for each treatment. The lamps were turned on for 72 h prior to the experimental start to stabilise the radiation. PAR was measured with Apogee Quantum Meter. Apollo spectroradiometer (SKA 400/UVB sensor, Skye Instruments LTD, Powys, UK) was used initially to adjust irradiation by changing the distance between the light bulbs and the table with cabbage leaves. Detailed measurements of UV intensity and radiation spectrum were performed with JAZ spectrometer (Ocean Optics, Dunedin, FL, USA). Leaves were exposed on either the ventral or dorsal side (blind/random). Lamps were placed at a distance of 65 cm above the table with leaves. For combined elicitor treatments, ultrasound treatment was performed prior to the radiation. The measured energy distribution of different wavelengths for the experimental radiation treatments is shown in Figure 1. A cellulose acetate filter (CA-foil, crystal clear, 0.14 mm × 650 mm × 1200 mm: art nr. I81.1.1-1706 L, Nordbergstekniska AB, Vallentuna, Sweden) was placed 4 cm above the leaf samples during the treatment to block fractions of UV-C emitted by the light bulbs. The UV lamps with UV-C filter used in our treatments emitted radiation in the wavelength area of 280–580 nm, with primary radiation at 280–350 nm and five narrow peaks in the area between 350 and 580 nm, with mercury emissions at 313–365 nm. The UV-radiation lamps had a prominent peak in the UV-B range at 280–315 nm and little radiation in the UV-A range of 320–400 nm (Figure 1).

**Figure 1 molecules-27-05256-f001:**
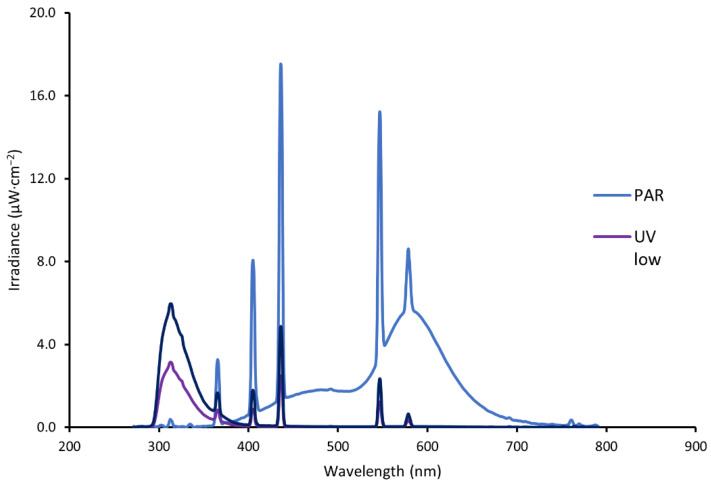
The plot shows spectral energy distribution in test chambers for the irradiation from TL-D 640 lamps giving mainly photosynthetic active radiation (PAR), and UV-B 313 El lamp (mainly ultraviolet radiation, UV-B and UV-A) at two levels: UV low (one lamp) and UV high (two lamps). Leaves were irradiated at a 65 cm distance with a UV-C filter 4 cm above the plants.

### 2.2. Sample Preparation 

After treatment, the leaves were sampled in random order and frozen in liquid nitrogen. Due to the time used for the preparation of samples (1 h in total), the final darkness period varied between 4 and 5 h. During exposure to liquid nitrogen, the leaves became brittle, and they were broken into parts (0 to 2 cm pieces) in a stone mortar to facilitate the preparation of representative samples of smaller volumes. The samples were vacuum-packed and stored at −80 °C until subsamples of frozen leaf pieces (50–60 g) were freeze-dried using a Biobase freeze dryer (BK-FD10S, Jinan, China) until a stable temperature over 12 h was reached (60 h in total). After lyophilisation, samples were individually packed in PE bags under vacuum and stored for 2 weeks at −80 °C before sending at ambient temperature to Gdansk University of Technology, Poland, for chemical analysis.

### 2.3. Chemical Analysis

#### 2.3.1. Chemicals

The following chemicals were used: methanol, acetonitrile, isopropanol (HPLC grade, Merck, Darmstadt, Germany), formic acid (>96.0%), 2,2-azinobis-(ethyl-2,3-dihydrobenzothiazoline-6-sulphonic acid) diammonium salt-ABTS, 6-hydroxy-2,5,7,8-tetramethylchroman-2-carboxylic acid-Trolox, caffeic acid, ferulic acid, sinapic acid, quercetin, kaempferol, indole-3-carbinol, indole-3-acetonitrile, 3,3′-diindolylmethane, *N*-acetyl-l-cysteine, allyl isothiocyanate, sulforaphane (Sigma Aldrich, St. Louis, MO, USA), and glucotropaeolin (AppliChem, Darmstadt, Germany).

#### 2.3.2. Determination of Phenolic Compounds

The phenolic compounds were determined in crude, methanolic extracts. A portion of 60 mg freeze-dried cabbage leaves was mixed with 1 mL of methanol (80%) and sonicated for 15 min. The samples were centrifuged (13,000 rpm, 5 min, 4 °C), and supernatants were collected. The extraction was repeated with a new portion of methanol (80%, 1 mL), and both obtained supernatants were pooled. The extracts were directly injected (20 µL) into the HPLC-DAD-MS system (1200 series, Agilent Technologies, Wilmington, DE, USA). A Kinetex XB-C18 100 Å column (150 mm length × 4.6 mm internal diameter, 5 μm particle size, Phenomenex) was used. The mobile phase contained 0.1% (*v*/*v*) formic acid in water (A) and 0.1% (*v*/*v*) formic acid in acetonitrile (B). The linear gradient applied was 10–100% B in 30 min at a flow rate of 1 mL∙min^−1^. Absorbance spectra were recorded between 190 and 700 nm. Chromatograms were monitored at 335 nm for hydroxycinnamates, at 360 nm for flavonols, and at 525 for anthocyanins. MS parameters were as follows: capillary voltage, 3000 V; fragmentor, 120 V; drying gas temperature, 350 °C; gas flow (N_2_), 12 L∙min^−1^; and nebuliser pressure, 35 psig. The instrument was operated both in positive and negative ion mode, scanning from *m*/*z* 100 to 1000. The peaks were tentatively identified by comparison of UV and MS spectra with literature data. The quantification of the analytes was performed with external calibration curves generated by the integration of the areas of absorption peaks of reference standards (caffeic, ferulic, or sinapic acid at 325 nm and flavonols as quercetin or kaempferol at 360 nm). The hydroxycinnamates were quantified respectively as caffeic, ferulic, or sinapic acid at 325 nm and flavonols as quercetin or kaempferol at 360 nm. For each sample, two parallel extractions and analyses were performed.

#### 2.3.3. Determination of Antioxidant Activity and Profiling of Antioxidants

The determination of the antioxidant activity of cabbage samples was carried out for the same extracts that were prepared for the determination of polyphenols. The profiling of antioxidants was performed by post-column derivatisation, as described earlier [23] with slight modifications. The post-column addition of ABTS derivatisation reagent (0.7 mmol/L in methanol) to HPLC eluate was done using a Pinnacle PCX instrument (Pickering Laboratories Inc., Mountain View, CA, USA). Derivatisation was carried out at 130 °C, with the flow rate of derivatisation reagent set at 0.1 mL∙min^−1^. Chromatograms of the products formed after derivatisation of antioxidant compounds with ABTS reagent were registered at 734 nm using a multiple-wavelength detector (1200 series, Agilent Technologies, Wilmington, DE, USA). The conditions of chromatographic separation were the same as in the case of the determination of polyphenols. The antioxidant activity of cabbage samples was expressed as a sum of Trolox equivalents (TE) calculated with the use of the equation of the Trolox calibration curve for areas under the negative peaks of analytes recorded in HPLC analyses during post-column derivatisation with ABTS reagent.

#### 2.3.4. Determination of Glucosinolates and Their Degradation Products

In the plant tissue, myrosinase and GLS are sequestered in separate cellular compartments. Upon plant tissue damage induced by pest attack or other disruption events, e.g., cutting or chewing, the initially separated substrate-enzyme defence system comes into contact, and GLS undergo hydrolysis. The first step is catalysed by myrosinase to the unstable intermediate, which rearranges further either spontaneously or upon the action of specifier proteins to isothiocyanates (ITC) and related products such as nitriles, epithionitriles, indoles, thiocyanates, or oxazolidine-2-thiones. In this study, the two most important groups of GLS hydrolysis products released by endogenous enzymes were determined: ITCs and indoles. The identity of each separated GLS confirmed by MS fragmentation pattern and characterised by major molecular ion peaks are seen in Table 2.

Determination of GLS and their degradation products was carried out in cabbage extracts with inactive or active endogenous myrosinase, respectively. The conversion rate of GLS to enzymatic degradation products was calculated separately for aliphatic and indolic GLS. For aliphatic GLS, the conversion rate to ITC was calculated according to the formula:CR_aliphatic_ = (ITC/GLS_aliphatic_) × 100(1)
where CR_aliphatic_: conversion rate of aliphatic GLS to isothiocyanates (%); ITC: total isothiocyanates content (µmol/g d.w.); and GLS_aliphatic_: total aliphatic GLS content (µmol/g d.w.).

In the case of indolic GLS for calculation of conversion rate to indoles, Equation (2) was used.
CR_indolic_ = (IND/GLS_indolic_) × 100(2)
where CR_indolic_: conversion rate of indolic GLS to indoles (%); IND: total indoles content (µmol/g d.w.); and GLS_indolic_: total indolic GLS content (µmol/g d.w.).

Determination of GLS was performed according to the standard ISO 9167 procedure with the modification described by Kusznierewicz et al. [24] for sample preparation and HPLC-DAD-MS analysis. The chromatographic peaks were first detected with DAD at 229 nm, then the identity of individual desulfo-glucosinolates (DS-GLS) was confirmed via API-ESI-MS (6130 Quadrupole, Agilent Technologies, Wilmington, DE, USA). The GLS level in each sample was quantified with the internal standard method using glucotropaeolin (GTL). For each sample, two parallel extractions and analyses were performed. The identity of each separated GL was confirmed based on MS fragmentation pattern characterised by major molecular ion peak, that in the case of positive ionisation occurs as sodium adduct [M_DS-GLS_ + Na]^+^ and in negative ionisation as a chloride adduct [M_DS-GLS_ + Cl]^−^.

Extraction and analysis of degradation products of GLS were performed according to procedures described earlier [25,26] with slight modifications. For each sample, two parallel extractions and analyses were performed. Briefly, to prepare extracts, 400 mg of freeze-dried cabbage leaves was mixed with 10 mL of phosphate buffer (0.01 M, pH = 7.4) and incubated for 3 h at 37 °C to enable conversion of GLS into isothiocyanates (ITCs) and indoles by endogenous myrosinase. Then, the samples were centrifuged (5000 rpm, 15 min) and supernatants collected and subjected to SPE procedure according to the method described by Pilipczuk et al. [26].

For the determination of indoles, the obtained SPE eluate was directly injected (30 µL) to the HPLC-DAD-FLD system (1200 series, Agilent Technologies, Wilmington, DE, USA) equipped with Kinetex PFP 100 Å column (150 × 4.6 mm, 5 μm, Phenomenex). The mobile phase consisted of 0.01% (*v*/*v*) formic acid in water (A) and 0.01% (*v*/*v*) formic acid in acetonitrile (B); the flow rate was set at 1 mL∙min^−1^. The mobile phase gradient was programmed as follows: Initially, 10% B increased linearly to 50% B over 15 min, then to 100 % B until 20 min. Indoles were monitored by UV detection at 280 nm and fluorescence detection at 280/360 (ex./em.). The contents of indolic compounds in different samples were calculated from the calibration curves generated for indole-3-carbinol, indole-3-acetonitrile, and 3,3′-diindolylmethane.

For ITC determination, 250 μL of SPE eluate was used for derivatisation with N-acetyl-L-cysteine and then HPLC-DAD-MS analysis according to the procedure described by Pilipczuk et al. [26]. The identification of dithiocarbamates was performed in positive ionisation modes by identifying the parent ion [M + H]^+^. The quantification of the analytes for which standards was available (allyl isothiocyanate, sulforaphane) was performed with external calibration curves, whereas for the other identified analyte (3-(methylsulfonyl)propyl isothiocyanate-3-MSP-ITC), the measured chromatographic area was substituted into the calibration equation of the reference standards with a similar structure (sulforaphane).

#### 2.3.5. Test of Possible Temperature Effect

The elicitor experiment was performed at 18 °C to facilitate rapid production of biochemicals by the plant cells and thereby conditions that could be relevant for the industry. Because of this, a possible temperature effect could be assumed, and therefore leaves directly from cold storage in darkness (4 ± 1 °C) were also analysed and compared to the reference sample from 18 °C in darkness. The results confirmed no temperature effect on the levels of phenolics and most of the glucosinolates. Only for neoglucobrassicin, which was a minor, dominant indolic glucosinolate in this crop, a 55% reduction was seen by increasing temperature from 4 to 18 °C (significant differences). Thus, it can be assumed that there was no interfering effect by temperature for all the other biochemicals investigated.

### 2.4. Experimental Design and Statistics

Leaves directly from cold storage in darkness (4 ± 1 °C) were compared to the reference sample from 18 °C in darkness, using one way ANOVA with temperature as fixed factor and repetition [3] as a random factor (Minitab 18). The experimental factors, namely irradiation (UV^high^, UV_low_, PAR) and ultrasound treatment (4 levels: 0-, 2-, 4- or 8-min duration), were tested on a reduced set of combinations to give eight different elicitor treatments and one untreated control, as described in Table 1. The treatment order and placement in radiation chambers were randomised. The experiment was run without parallels, but the entire experiment (9 treatment combinations) was repeated three times continuously over a period of 3.5 days, giving *n* = 9, repetitions = 3 and total samples = 27. Differences between treatments and treatment combinations were analysed by ANOVA ≤ 0.05 (Minitab 18) for 19 response variables: specific polyphenols, glucosinolates and degradation products, antioxidant activity, and dry matter. Analysis was performed with treatment as a fixed factor (9 levels) and ‘experimental repetition in successive periods’ (3 levels) as a random factor. For results with significant differences by ANOVA, differences between treatments were investigated by Tukey’s test at *p* ≤ 0.05. Correlation analyses (Minitab 18) were performed between chemical constituents and exposure to 1) UV irradiation or 2) PAR or total irradiation energy.

To find the effective elicitors concerning phytochemicals increase in Brassica leaves, the following statistical hypotheses were tested for specific compounds of phytochemicals: H1: Phytochemicals was higher in leaves treated with ultrasound (15–60 kJ∙l^−1^) than the non-treated control. H2: Phytochemicals were higher in leaves treated with PAR irradiation (497 kJ∙m^−2^) than in non-treated control. H3: Phytochemicals were higher in leaves treated with UV radiation (59 to 99 kJ∙m^−2^) than in the untreated control. H1, H2, and H3 were tested against the alternative hypothesis H0: Treatments have no influence or adverse effect on the content of specific phytochemicals. For elicitors and compounds where H1, H2, or H3 were confirmed, the following hypotheses were tested: H4: Phytochemicals increase with an increase of elicitor level (2 levels of UV, three levels of ultrasound). H5: Phytochemicals were stimulated to a different extent by the different elicitor factors. H6: Phytochemicals increase additionally by combining 2 or more of the elicitor factors. H4, H5, and H6 were tested against alternative hypothesis, H0: No differences between elicitor factors or no effect in the level of phytochemicals by increasing the elicitor level or combining elicitor factors. 

## 3. Results

The time-related factor for the three ‘experimental repetition in successive periods’ showed no significant differences for any of the tested response variables for the three experiment repetitions over 3.5 days. Thus, it can be concluded that the time delay between subsequent trial repetitions had no influence on results.

### 3.1. Phenolic Compounds

The hydroxycinnamates were quantified respectively as caffeic, ferulic, or sinapic acid, and flavonols as quercetin or kaempferol (Table 2). Example chromatograms of crude cabbage extracts are seen in Figure 2A,B.

Elicitor-treatments with UV and the treatment with PAR had positive effects on several of the detected phenolic compounds (Table 2), resulting in higher concentrations of flavonols, hydroxycinnamates, and anthocyanins, compared to the untreated control (Figure 3A,B). Total radiation given by the different treatments (Figure 1) was correlated with total flavonols (R = 0.82) and anthocyanins (R = 0.96), all at P < 0.001. In the non-treated control samples, there was notably no prior detection of any flavonols and anthocyanins, and only three of seven hydroxycinnamates were present (sinapoyl-hexocide, disinapoyl-gentiobioside, and trisinapoyl-gentiobioside). Of these, sin-hex were affected by UV treatment, while the latter two were not significantly influenced by any of the elicitor treatments.

There were some differences among the phenolic classes in response to either PAR or UV radiation. UV treatment gave higher levels than PAR for the flavonols quercetin-3-diglucoside-7-glucoside (2.3-fold higher) and kaempferol-3-diglucoside-7-glucoside (3.0-fold higher) (Figure 3A). When PAR + UV combinations were compared to PAR alone, considerably higher concentrations were seen for hydroxycinnamates quercetin 3-diglucoside-7-glucoside (4.5-fold higher), kampferol 3-diglucoside-7-glucoside (4.2-fold higher), Q-3-caffeoyl-hexoside (10-fold higher), coumaroyl-hexoside (6.4-fold higher), coumaroyl-hexoside-1 (8.0-fold higher), feruloyl-hexoside (3.5-fold higher), and sinapoyl-hexocide (1.6-fold higher). The five first mentioned compounds were also significantly higher when PAR+UV combinations were compared with UV alone (When PAR + UV combinations were compared 2-fold higher, 1.4, 1.4, 1.6 and 1.7-fold higher, respectively). Anthocyanins, in contrast, had significantly higher concentrations when comparing PAR exposed leaves (2 h pre-treatment) with the treatment with UV given to dark exposed leaves (Figure 3A,B). For the affected flavonols and hydroxycinnamic acids, there also appeared to be a positive interaction between UV + PAR as leaves were producing higher concentrations when these two were combined in contrast to either UV or PAR. However, there was no significant effect on any of these compounds by increasing UV irradiance energy from low to high level (59 versus 99 kJ∙m^−2^). During the experiment, a purple colour appeared on the non-exposed side of the leaves in the UV treatment (Appendix A). In contrast, on leaves stored in PAR without UV, the purple colour appeared on the exposed side of the leaves.

Elicitor treatments with sonication had no significant effect on concentrations of any of the detected phenolic compounds. Furthermore, sonication treatment in combination with PAR and UV did not cause any further increase in the concentration of flavonols, hydroxycinnamates, or anthocyanins. 

### 3.2. Glucosinolates and Their Hydrolysis Products

The content and composition of GLS in cabbage leaves were determined for their desulfo-derivatives (DS-GLS). The chromatographic profile of GLS for cabbage studied revealed the presence of six major GLS compounds (Figure 4A). Four of them belonged to the aliphatic GLS, and the other two to indolic. The predominant GLS in cabbage studied was sinigrin (Figure 4D). None of the elicitor treatments of PAR, UV radiation, or ultrasonic treatment had a significant effect on any of the quantified aliphatic or indolic glucosinolates (Figure 4B,E). The aliphatic GLS sum up to a total concentration of around 10 µmol∙g^−1^ DM, while indolic GLS (glucobrassicin and neo-glucobrassicin) sum up to a total level below 0.5 µmol∙g^−1^ DM. The most abundant ITC detected in the studied material was 3-(methylsulphinyl)propyl isothiocyanate (2.1 to 2.6 µmol∙g^−1^ d.w.) (Figure 4E).

Elicitor treatments also did not have any significant effect on breakdown products of the total aliphatic GLS (Figure 5A), total isothiocyanates (Figure 5B), aliphatic GLS conversion rate to ITC (Figure 5C), or total indolic GLS (Figure 5D). For total indole content, there were some inconsistencies with minor differences between two of the treatments (Figure 5E). The total content of ITCs in cabbage leaves was about 3.15 µmol∙g^−1^ d.w. (Figure 5B). The conversion rate of aliphatic GLS to corresponding ITCs reached nearly 32% (Figure 5C), and the indolic conversion rate was 9–13% (Figure 5F).

No effect of elicitors was seen for the glucosinolate precursors, isothiocyanates (Figure 4B,E). However, as a result of indolic GLS degradation, three major compounds were detected: indole-3-acetonitrile, 3,3’-diindolylmethane, and indole-3-carbinol (Figure 4C,F). Only the content of one of them, the indole-3-acetonitrile, was significantly affected by elicitor treatment with a 38% reduction (compared to untreated control) for the treatment combinations with UV + PAR (Figure 4F). The two latter indoles were not possible to separate by multiple comparison tests between treatments, even though ANOVA showed P < 0.05 due to a considerable variation within PAR treatments when compared with non-irradiated samples. Total indoles tended to decrease in response to UV radiation as indicated by a clear trend for the four treatments including UV (Figure 5E), even though the effect was statistically significant only between ‘PAR + UV^high^ + S_8_’ and ‘S_2_’. When it comes to the specific indoles (Figure 4F), a 1.6- and a 1.8-fold decrease was seen in indole-3-acetonitrile (I3ACN) at low and high UV radiation level, respectively. For diindolylmethane (DIM), a 1.7-fold decrease was observed at high UV radiation level.

There was no significant difference according to Tukey’s multiple comparison test between any of the treatments in aliphatic GLS conversion nor in indolic GLS.

### 3.3. Antioxidant Activity

The total antioxidant activity of cabbage leaves in the study was determined along with the composition of antioxidants by HPLC post-column derivatisation with ABTS reagent. In Figure 2, the chromatograms of separated analytes (upper chromatograms at 335 nm) were set with chromatograms obtained after post-column derivatisation with ABTS regent (bottom chromatograms at 734 nm). The detection of the solution bleaching as a result of the reduction of ABTS radicals corresponding to the antioxidant activity of analytes in post-column effluent is reflected by the ‘negative peak’ in chromatograms at 734 nm. The presence of ‘negative peaks’ on a chromatogram registered after derivatisation revealed which individual components were responsible for the antioxidative potential of a sample analysed and to what extent. The obtained antioxidant profiles indicated that the primary antioxidants in cabbage samples studied are phytochemicals present in unresolved polar fraction visible at the start of the chromatogram (Figure 2A,B, bottom chromatograms, peak marked *). Besides some hydroxycinnamic acid derivatives (Figure 2, peaks 7, 8 and 9), also present in untreated leaves (A), the irradiated samples (B) contained the additional newly generated antioxidant, quercetin triglucoside (Figure 2B, peak 2). The presence of this new antioxidant in irradiated cabbage leaves increased the total antioxidant activity in these samples by about 20–40%. The total antioxidant activity of cabbage samples was assessed as a sum of areas under negative peaks of identified groups of phytochemicals, calculated as Trolox equivalents and presented as a bar graph (Figure 3B). The total antioxidant activity of all cabbage samples studied varied from 11.7 to 16.9 µmol∙TE∙g^−1^ d.w.

The results conclude that PAR treatment of leaves had no effect on antioxidant activity (AA-TE) of leaves while UV had a moderate effect (Figure 3B). However, when PAR was given in combinations with UV radiation, total antioxidant activity was significantly higher than UV alone (16%) or PAR alone (38%). There was no apparent interaction between sonication, together with PAR or UV radiation. Elicitor treatments with sonication alone did not have any significant effect on total antioxidant activity.

## 4. Discussion

### 4.1. Phenolic Compounds

The investigation of the stimulating effect by fluorescent UV and PAR on flavonols, hydroxycinnamates, and anthocyanins is in accordance with results from other studies [12,27,28,29,30]. A similar change pattern in phenolic composition following storage and UV treatment was observed by Harbaum-Piayda et al. [12]. They also reported an increase in the content of sinapoyl hexoside after cabbage storage for 4 days in the dark by about 60% and under UV radiation by nearly 300%. The increase in hydroxycinnamic acids in our study is in accordance with the results by Duarte-Sierra [9] on broccoli, but the increase, after a quite comparable UV exposure, was much higher (3-fold increase after 19 h v.s. 10–15% increase in their study for broccoli after 3–14 days). Thus, the leaves of head cabbage seem to react more strongly to the UV treatment than broccoli florets. The difference can also be explained by differences in radiation spectrum, as the tested UVB-313 EL lamps contained not only UV-B but also some UV-A, while the broccoli study used only UV-B (Figure 1 and Table 1). The concentrations of newly generated phenolics reported by Harbaum-Piayda et al. [12] are six times higher than in our studies. This can be due to different durations of UV exposure: 4 days versus 10 h in our study. With respect to time given after treatment for the biosynthesis of compounds, a 4–5 h period in our experiment seems to be enough to facilitate the biosynthesis of individual phenolic compounds. This is in accordance with results by Darré et al. [31], which found a peak of phenolics 6 h after UV-exposure in a study with broccoli. The increase in the content of phenolic compounds, induced in our study by PAR (497 kJ∙m^−2^), was also seen in the results reported for *Brassica campestris* [27], performed with PAR levels 6 to 9 times higher than in our study and 33–51 times longer duration of exposure (25–35 d versus 18 h in our study). In their study, the production of bioactive compounds was 10–20 times higher than in our studies. This could be explained by the higher elicitation stimuli, longer exposure time, and use of intact, still-growing plants, where a more efficient biosynthesis of secondary metabolites could be expected. They also tested a different Brassica species than we did, which may have a different capacity for the biosynthesis of compounds. The calculated contents of total flavonols determined in our UV exposed samples (range 0.15–0.40 mg∙g^−1^ d.w.) were lower than those previously identified (0.7–6.0∙mg g^−1^ d.w.) for cabbage cultivated and treated in Research Institute of Horticulture in Skierniewice, in Poland (Kusznierewicz, B. et al., 2017, Poster 34, Proceedings, 4th International Glucosinolate Conference, 2017). In this former case, the treated leaves were outer and dark green as opposed to our pale inner leaves of harvested and cold-stored cabbages. Thus, the pale inner leaves were devoid of flavonols initially, while outer green cabbage leaves contained these compounds before experiments. Since the outer green leaves were exposed to sunlight during growth, the phenylpropanoid pathway could have been stimulated already preharvest. 

Our results of the UV-induced increase in anthocyanin levels and purple colouring of leaves are in accordance with the appearance of anthocyanins in UV-exposed *Arabidopsis* plants observed in the study by Ries et al. [30]. Our results may indicate a more potent effect of PAR than UV radiation on the increase of anthocyanins within the given biosynthesis time. This agrees with a study reported by Cominelli et al. [29] and by Lercari et al. [32], where PAR exposure had a substantial effect on the increase of anthocyanins, respectively, in *Arabidopsis thaliana* or white cabbage. 

Since the results indicate that sonication had no effect on the measured constituents, it can be assumed that sonication at the given doses neither improve nor hinder any inducer or enhancer properties of UV radiation. The lack of impact of ultrasound treatment in our study was in contrast with findings on Romaine lettuce leaves presented by Yu et al. [21], where an increased phenylalanine ammonia lyase (PAL) activity was observed, leading to the increased production of phenolic compounds. Compared to the lettuce study, the present study was performed with a higher intensity of ultrasound treatment and longer exposure time (2–8 min of 35 kHz, 128 W∙l^−1^ versus 1–3 min of 25 kHz, 26 W∙l^−1^). The differences in PAL are most likely to be due to different species, which may differ in capacity for biosynthesis of compounds. Also, differences in leaf morphology could add to this explanation as *Brassica* leaves are much thicker with a thick wax layer and more robust than lettuce leaves, so it could be assumed that thicker leaves need a more substantial exposure to respond to the treatment. A shorter storage time after exposure in our study could be another reason for the observed lack of effect (4–5 h versus 30–90 h for the lettuce study). Most likely, plant type and the time necessary to trigger biosynthesis could explain the differences between the two studies. In the study by Yu et al. [21], the storage time after treatment was a very crucial factor, as there was an interplay between factors that promoted the production of phenolics and those that consumed them. After 30 h of storage, the consumption of total phenolics seemed to surpass the accumulation since the level was lower than the initial levels in the study. 

The profile and content of phenols in white cabbage may also vary depending on the variety and growing conditions [33,34]. Therefore, as indicated by the obtained results, regardless of the initial quality of the vegetable raw material, post-harvest treatment of cabbage with UV can induce the formation of some phenolic compounds as well as enhance their biosynthesis.

### 4.2. Glucosinolates and Their Autolysis Products

The lack of influence of UV on all GLS in our study is in accordance with post-harvest UV radiation, not altering total aliphatic GLS levels in broccoli microgreens [14]. In contrast to our study, Duarte-Sierra et al. [9] found the total glucobrassicins to increase by 18–22% by UV-B radiation, similar to our dose, albeit with longer storage time before sampling. Similarly, in another study [35] with storage of broccoli, PAR (1.6 times our radiation energy) + UV (similar to our level) increased the level of total glucosinolates, total indolyl and aliphatic GLS, and some individual glucosinolates. However, Darré et al. [31] found contrasting results showing that total GLS decreased in response to UV treatment. At the same time, aliphatic GLS, and especially glucoraphanin content, were increased by high-intensity, short exposure of UV radiation (3 or12 kJ∙m^−2^, 12 min exposure time +18 h in darkness) of newly harvested broccoli heads. More studies need to be performed to understand the effect of GLS in Brassica species.

In a study by Harbaum-Piayda et al. [12], the content of glucobrassicin and 4-methoxyglucobrassicin in the UV-B treated plant material dropped approximately from 0.3 to 0.1 mg∙g^−1^ d.w. and from 0.27 to 0.05 mg∙g^−1^ d.w., respectively. This was not seen for the same individual glucosinolates in our study but for the indoles. The drop in indoles can be explained in at least three different, not mutually exclusive, ways. Firstly, the indolic compounds are light sensitive [36], and UV irradiation may cause their degradation. It is possible that this sensitivity is not the same for indolic GLS as for their degradation products and that indoles are possibly more light-sensitive than parent compounds. Secondly, differences in levels of main degradation products can result from the formation of ascorbigen and dihydromethoxyascorbigen, which may be more efficient in leaves treated with UV. These compounds, not detected in our analytical system, are formed in the reaction of indole-3-carbinol with ascorbic acid. Harbaum-Piayda et al. [12] reported the increase of ascorbigen and dihydromethoxy-ascorbigen levels in cabbage leaves treated post-harvest with UV-B. Finally, the lower content of products of indolic GLS autolysis in UV-treated leaves can also result from changes in endogenous myrosinase activity. According to Reifenrath & Müller [37], GLS and myrosinase, both partners of the defence system of *Brassicaceae*, responded highly species-specific to UV exposure. 

The predominant GLS in cabbage in our study was sinigrin, which represented about 70% of total GLS (6.9 to 7.8 µmol∙g^−1^ d.w.). This is similar to studies by Kołodziejski et al. [38,39,40], where sinigrin accounted for 62%, 55% and 44% of total GLS, respectively. In other studies with white cabbage leaves, this value oscillated approximately between 1.2 µmol∙g^−1^ d.w. and 11 µmol∙g^−1^ d.w. [39,40,41]. This is similar to 10 µmol∙g^−1^ d.w. total aliphatic GLS found in our study.

According to [42], the conversion rate of aliphatic GLS to corresponding ITCs was 17% in green cauliflower and 84% in purple cauliflower. However, indolic GLS showed an inverse relationship and were formed in these plants with a rate of 80 and 27%, respectively. This is higher than the observed indolic conversion rate in our study (9 to 13%).

Finding the ITC compound 3-(methylsulphinyl)propyl isothiocyanate as the most abundant ITC in our study (2.1 to 2.6 µmol∙g^−1^ d.w.) is confirmed in studies by Pilipczuk et al. [26] and Koss-Mikołajczyk et al. [39], which also determined this compound as a major ITC in white cabbage extracts after GLS autolysis. Its content in those studies amounted to 3.80 µmol∙g^−1^ d.w. and 0.30 µmol∙g^−1^ d.w., respectively.

### 4.3. Antioxidant Activity

In our study, only treatment of UV radiation given in combination with PAR increased the antioxidant activity. These samples had coinciding increased levels of six individual phenolics, which makes it tempting to interpret some of these to be sources for the observed increase in antioxidant activity. Darre et al. [31] found UV exposure (3.2 to 5.0 W∙m^−2^) as an appropriate treatment to increase the antioxidant activity in broccoli after harvest. Their antioxidant activity peaked 2 and 6 h after treatment for the low and high-intensity irradiation, respectively, which was similar to the post-treatment in our study (4–5 h). The total antioxidant activity of our cabbage samples studied varied from 11.7 to 16.9 µmol∙TE∙g^−1^ d.w., which is consistent with the data reported in the literature [39]. 

Exposure to UV appeared to stimulate de novo formation of flavonols and some hydroxycinnamate monosaccharides, while PAR had a more moderate effect. For anthocyanins, in contrast, PAR had a significantly stronger effect than UV exposure alone. The ultrasound treatment had no effect on the content of any phytochemicals or antioxidant activity in our study, and glucosinolates were largely unaffected by any of the elicitor treatments. The combination of UV and PAR may thus be used to increase phenolics’ content and thereby increase the value of leafy waste fractions of white cabbage processing. Our results demonstrate that treatment with UV at 59 kJ m^−2^ in combination with PAR at 497 kJ∙m^−2^) may substantially improve the content of phenolics and antioxidant activity with no adverse effect on main GLS, except for a reduction in indole-3-acetonitrile. Treatment with ultrasound at 35 kHz (128 W∙l^−1^ water) for up to 8 min duration (15–61 kJ∙m^−2^) did not affect the content of phytochemicals. The increase in the content of four phenolics and antioxidant activity by combing UV+PAR indicates a possible overlap in the signal transduction pathways of cabbage leaves’ responses to UV and PAR. However, since the small amounts of unwanted radiation were not filtered from lamps to have precise treatments of only PAR or UV, such results cannot with certainty be concluded from our study.

The results revealed hypothesis H1 to be rejected, as the ultrasound at 15–61 kJ∙l^−1^ water was not sufficient to increase the nutraceutical value of the Brassica rest raw material. Hypotheses H2 and H3 of increasing biochemical content were confirmed for PAR (497 kJ∙m^−2^) and UV (59–99 kJ∙m^−2^) for the content of phenolics and antioxidant activity, but not for GLS and their degradation products. H4 was rejected as there was no increase of phenolics or other compounds by increasing UV irradiation intensity from 59 to 99 kJ∙m^−2^. H5 was true for anthocyanins and PAR versus UV treatment, which shows PAR three times as effective as UV. H6 was confirmed true for the combination of PAR + UV for the content of five phenolics, hydroxycinnamic monosaccharides, and antioxidant activity, but not for GLS and their degradation products. 

## 5. Conclusions

The combination of PAR at 497 kJ∙m^−2^ + UV at 59 kJ∙m^−2^ was found to be the most effective elicitor combination by significantly increasing the content of phenolics and hydroxycinnamate monosaccharides by a doubling to 10-fold increase, as well as enhancing antioxidant activity by 30% in the Brassica leaf rest fraction. Ultrasound at 15–60 kJ∙l^−1^ did not affect the phytochemicals. Based on the promising results on lettuce found in the literature, higher energy levels of ultrasound should be tested in follow-up studies before making conclusions on possible elicitor effects of ultrasound on phytochemicals in cabbage leaves. GLS were not affected by UV, PAR, or ultrasound, except for total indoles, which tended to decrease by UV treatment (59 kJ∙m^−2^). The tested abiotic elicitors seem to be a less relevant tool for increasing GLS content in cabbage rest raw materials. Nevertheless, the GLS content does not appear to be reduced when such treatment is used to increase other phytochemicals. To address the economic justification of such elicitor treatments in the food industry, further studies should investigate methods for repeated exposure of the rest raw materials over the required time-period for elicitation, for implementation in large scale equipment innovations like assembling bands or rotating lattice drum systems.

## Figures and Tables

**Figure 2 molecules-27-05256-f002:**
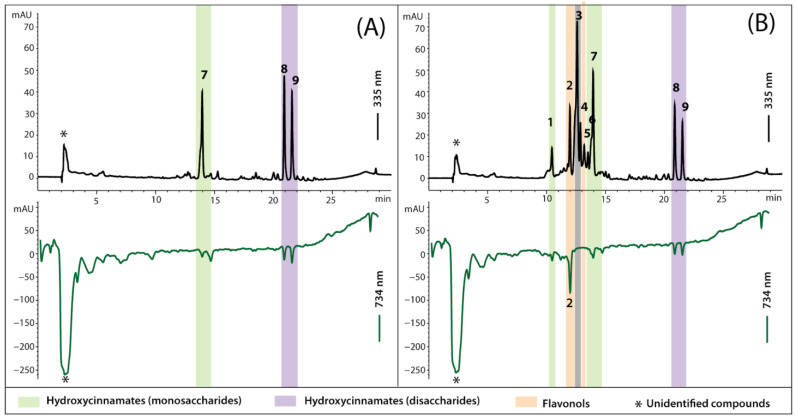
Chromatograms of crude cabbage extracts with (**A**) the lowest (non-treated leaves from cold storage in darkness) and (**B**) the highest (UV^high^ in PAR) content of phenolics, detected at 335 nm set with profiles of antioxidants detected on-line after derivatisation with ABTS reagent and traced at 734 nm. The identity of peaks is given in Table 2.

**Figure 3 molecules-27-05256-f003:**
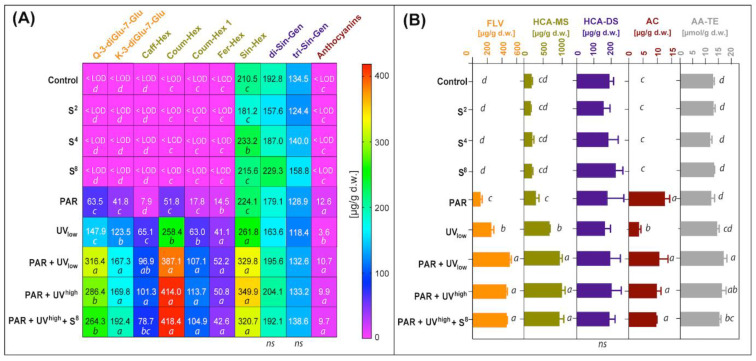
The contents of (**A**) major individual phenolic compound combined with (**B**) the content of total flavonols (FLV), hydroxycinnamate monosaccharides (HCA-MS), hydroxycinnamate disaccharides (HCA-DS), anthocyanins (AC), and total antioxidant activity expressed as Trolox equivalents (AA-TE) determined in treated and untreated white cabbage samples. Treatment codes are given in Table 1. The names of analytes are abbreviated as follows: Q-3-diGlu-7-Glu, quercetin 3-diglucoside-7-glucoside; K-3-diGlu-7-Glu, kaempferol 3-diglucoside-7-glucoside; Caff-Hex, caffeoyl-hexoside; Coum-Hex, coumaroyl-hexoside; Fer-Hex, feruloyl-hexoside; Sin-Hex, sinapoyl-hexoside; di-Sin-Gen, disinapoyl-gentiobioside; tri-Sin-Gen, trisinapoyl-gentiobioside, LOD, limit of detection. The results marked as *ns*, not significant, refer to a comparison of treatments within each column by ANOVA (P ≤ 0.05). Numbers followed by the same letters within one column/parameter are not significantly different according to Tukey’s multiple comparison test at P ≤ 0.05.

**Figure 4 molecules-27-05256-f004:**
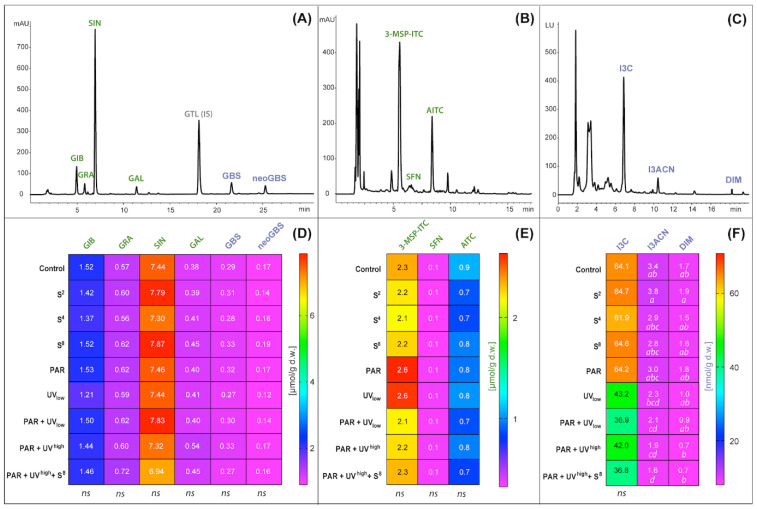
Sample chromatograms of (**A**) desulfo-glucosinolates (λ = 229 nm), (**B**) ITC-NAC conjugates (λ = 272 nm), and (**C**) indolic compounds (ex./em.: 280/360 nm) obtained from HPLC analysis of untreated reference white cabbage leaves (Ref), combined with the contents of (**D**) individual glucosinolates, (**E**) isothiocyanate, and (**F**) indole determined in treated and untreated white cabbage samples. Treatment codes are given in Table 1. The name of analytes are abbreviated as follows: GIB, glucoiberin; GRA, glucoraphanin; SIN, sinigrin; GAL, glucoalyssin; GTL(IS), glucotropaeolin (internal standard); GBS, glucobrassicin; neoGBS, neoglucobrassicin; 3-MSP-ITC, 3-(methylsulphinyl)propyl isothiocyanate; SFN, sulforaphane; AITC, allyl isothiocyanate; I3C, indole-3-carbinol; I3ACN indole-3-acetonitrile; DIM 3,3′-diindolylmethane. The results marked as *ns*, not significant, refer to a comparison of the parameter given in this column by ANOVA for all treatments and control (P ≤ 0.05). Numbers followed by the same letters within one column/parameter are not significantly different according to Tukey’s multiple comparison test at P ≤ 0.05.

**Figure 5 molecules-27-05256-f005:**
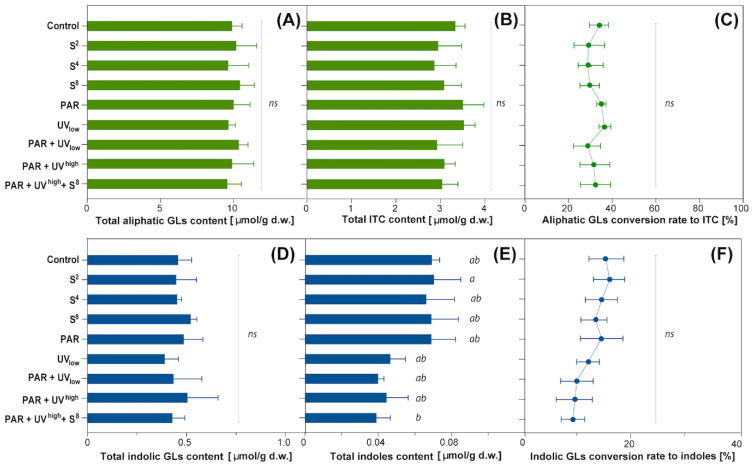
The contents of total aliphatic (**A**) and total indolic (**D**) GLS and the contents of corresponding total ITC (**B**) and total indoles (**E**) formed after autolysis of GLS were calculated as means +/− SD based on determinations for samples of treated and untreated white cabbage leaves derived from 3 independent experiments analysed in duplicates. The percentiles of conversion rates of GLS to respective breakdown products (**C**,**F**) were calculated for each pair separately, then means +/− SD were obtained for a given treatment. For treatment codes, see Table 1. The results marked as *ns* refer to no significant differences between treatments within the column/compound according to the analysis of variance at P ≤ 0.05. Numbers followed by the same letters within one graph are not significantly different according to Tukey’s multiple comparison test at P ≤ 0.05.

**Table 1 molecules-27-05256-t001:** Four-step elicitor treatment of white cabbage waste fraction leaves in comparison with control (non-treated). Leaves were exposed at a 65 cm distance from the irradiation source with a UV-C filter between lamps and the leaves. Samples were exposed (for min or h) or non-exposed (-) to one or more of the elicitors steps as follows: step (1) 2 h irradiation treatment; step (2) sonication for 2, 4 or 8 min; step (3) 10 h main irradiation; and step (4) 2 h of post-treatment irradiation. A final period of 4–5 h in darkness was given prior to sampling for analysis (total 18–19 h). Treatment codes: S = ultrasound sonication; PAR = photosynthetic active radiation (irradiation from TLD-840 lamps); UV = ultraviolet irradiation (from UVB-313 EL lamps, at low or high level). The irradiation spectral distribution is shown in Figure 1.

Treatment Codes	Pre-Treatment	2nd Treatm.	Main Irradiation Treatment	Post-Treatment		Irradiation Energy (kJ m^−2^)			Sonic Energy
	Radiation	Sonication		Radiation		UVB + UVA	PAR	Total	(kJ∙l^−1^ Water)
	TLD-840 *	DL510H ^¤^	TLD-840	UVB-313 EL	TLD-840	250–400 nm	400–700 nm	250–700 nm	35kHz
**Control**	-	-	-	-	-	0	0	0	0
**S^2^**	-	2 min	-	-	-	0	0	0	15
**S^4^**	-	4 min	-	-	-	0	0	0	31
**S^8^**	-	8 min	-	-	-	0	0	0	61
**PAR**	2 h	-	10 h	-	2 h	17	497	514	0
**UV_low_**	-	-	-	10 h	-	59	18	77	0
**PAR + UV_low_** ******	2 h	-	Both 10 h	Both 10 h	2 h	59	435	494	0
**PAR + UV^high^** *******	2 h	-	Both 10 h	Both 10 h	2 h	99	486	585	0
**PAR + UV^high^** ***** + S8**	2 h	8 min	-	10 h	2 h	99	486	585	61

^¤^ Ultrasound sonication, DL510H DIGIPLUS-Sonorex, 35 kHZ, 128 W∙l^−1^ water in sonication bath. * TLD-840 lamp: irrandiance 0.31 to 0.33 W∙m^−2^ (UV, UV, 250–400 nm) + 8.4 to 9.9 W∙m^−2^ (PAR 400–700 nm). ** UVB-131 EL lamp (UV low): irradiance 1.2 to 1.7 W∙m^−2^ (UV, 250–400 nm) + 0.3 to 0.5 W∙m^−2^ (PAR, 400–700 nm). *** UVB-131 EL lamp (UV high): irradiance: 2.3 W∙m^−2^ (UV, 250–400 nm) + 0.5 W∙m^−2^ (PAR, 400–700 nm).

**Table 2 molecules-27-05256-t002:** Chromatographic and spectrometric data obtained during HPLC-DAD-MS analyses of cabbage leave extracts used for putative identification of major secondary metabolites. Putative identification is given for all peaks numbered from 1–9 for phenolics and coded by letters for the desulfo-glucosinolates (DS-GLS) and their degradation products (isothiocyanates).

Peak	t_R_ [min]	λ_max_ (nm)	MS (+)	MS (−)	Mw	Putative Identification
1	10.6	300sh, 330	365 [M + Na]^+^	341 [M − H]^−^	342	Caffeoyl hexoside
2	12.2	255, 350	811 [M + Na]^+^	787 [M − H]^−^	788	Quercetin 3-diglucoside-7-glucoside
3	12.6	315	349 [M + Na]^+^	325 [M − H]^−^	326	Coumaroyl hexoside
4	12.9	266, 345	795 [M +Na]^+^	771 [M − H]^−^	772	Kaempferol 3-diglucoside-7-glucoside
5	13.2	315	349 [M + Na]^+^	325 [M − H]^−^	326	Coumaroyl hexoside
6	13.5	300sh, 330	379 [M + Na]^+^	355 [M − H]^−^	356	Feruloyl hexoside
7	13.9	330	409 [M + Na]^+^	385 [M − H]^−^	386	Sinapoyl hexoside
8	20.9	330	-	753 [M − H]^−^	754	Disinapoylgentiobioside
9	21.6	330	-	959 [M − H]^−^	960	Trisinapoylgentiobioside
Desulfoglucosinolates (DS-GLS)						
GIB	4.9	229	366 [M + Na]^+^	378 [M + Cl]^−^	343	Desulfo-glucoiberin
GRA	5.8	229	380 [M + Na]^+^	392 [M + Cl]^−^	357	Desulfo-glucoraphanin
SIN	6.9	229	302 [M + Na]^+^	314 [M + Cl]^−^	279	Desulfo-sinigrin
GAL	11.4	229	394 [M + Na]^+^	406 [M + Cl]^−^	371	Desulfo-glucoalyssin
GBS	21.7	220, 280	391 [M + Na]^+^	403 [M + Cl]^−^	368	Desulfo-glucobrassicin
neoGBS	25.4	220, 280	421 [M + Na]^+^	433 [M + Cl]^−^	398	Desulfo-neoglucobrassicin
Conjugates of Isothiocyanates with *N*-acetyl-l-cysteine (ITC-NAC)						
3-MSP-ITC	5.5	216, 250, 270	327 [M + H]^+^	-	326	3-(Methylsulphinyl)propyl-ITC-NAC
SFN	6.4	216, 250, 270	341 [M + H]^+^	-	340	Sulforaphane-NAC
AITC	8.4	216, 250, 270	263 [M + H]^+^	-	262	Allyl-ITC-NAC

## Data Availability

Not applicable.

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
