# Peer review of "Effects of Post-Harvest Elicitor Treatments with Ultrasound, UV- and Photosynthetic Active Radiation on Polyphenols, Glucosinolates and Antioxidant Activity in a Waste Fraction of White Cabbage (Brassica oleracea var. capitata)"

_molecules, 2022, doi:10.3390/molecules27165256_

Round 1

Reviewer 1 Report

The Authors investigate the impact of part harvest treatment (sonication, UV and PAR) on the accumulation of various classes of phytochemicals (phenolics, flavonoids, anthocyanins, glucosinolates) and antioxidant activity (using the in vitro ABTS assay) in the white cabbage Brassica oleracea var. capitata by-product.

The subject is of interest. I consider that the present results provide new information of interest for the readership of Molecules. The paper is well prepared.

I have only very minor suggestions to improve this work:

1.     Table 2 is cut on its right part (however no problem to understand it).

2.     Line 668: 1-fold is not an increase. Maybe rephrase this part. You observed upto 10-fold increase.

Author Response

Comment 1: 'Table 2, right column is cut': Last column in Table 2 is changed from font 10 to font 9 in last to give space for the full names of all compounds.

Comment 2: '1-fold' increase is not an increase'. One fold increase is in fact a doubling of content. (Example: Onefold of a paper doubles the thickness from one sheet to two). Nevertheless, for clarification, we have hanged: as follows in line 668: from ... 'by 1 to 10-fold increase' to ....'by a doubling to tenfold increase'   

Reviewer 2 Report

The manuscript concerned an interesting issue of investigating the effect of elicitor treatments on increasing the value of cabbage waste.

I have some comments on the manuscript.

1.       Phrases such as "we", "our" are rather not preferred in scientific publications. Please correct throughout the manuscript.

2.       Line 87-98. It is worth emphasizing the scientific novelty of the manuscript.

3.       Are the processes of improving the quality of agro-waste described in the manuscript economically justified in the context of application in the food industry?

4.       Has the "conclusion" section been omitted?

Author Response

Comments to reviewer 2: Line numbers are referring to the original numbers in the version given by the reviewers. In the original text line numbers are slightly moved due to the editing of text.

Comment 1:  Phrases such as "we", "our" are rather not preferred in scientific publications. Please correct throughout the manuscript.

  • > The use of active voice (we, our) is usually acknowledged as equal to passive voice by most journals. Nevertheless, the text is changed to passive voice by replacing the term 'we' during the entire manuscript as described below. The term 'Our' is also replaced by passive form during the manuscript except for the discussion part in cases where we suggest to use the term 'our' to be the term which most clearly separate the results representing the present study from the results from other studies. We are willing to change also this part if the editor still think this is wanted. 'Our' can in such case be replaced with 'the present study' or similar during the discussion part of the paper. 

Below is the list of changes we have performed to omit the 'we' and 'our' form (give a passive voice to the paper):

Line 13: We exposed A leaf waste fraction from industrial trimming of head cabbage was exposed to UV radiation ...

Line 89: Our The present study is an approach to reducing losses in the food chain 

Line 91: We selected uUltrasound was selected as an appropriate method based on its ability to enrich nutraceuticals in lettuce leaves within a very short elicitation period (21)

Line 96: The biochemicals focused in our the present study, phenols, GLS, isothiocyanates, and indoles, are among ...

Line 137: Since lettuce has tender leaves, we increased the dose was increased from 25 to 35 kHz due to the thicker leaves of cabbage.

Line 135: Since studies on ultrasound treatment of cabbage leaves are limited, exposure levels in our the present study were based on effective doses given for lettuce [21].

Line 180: The UV lamps with UV-C filter used in our the treatments emitted radiation in the wavelength....

Line 325: We aimed to perform t The elicitor experiment was performed at 18°C to facilitate rapid production of biochemicals by the plant cells and thereby conditions that could be relevant for the industry. 

Line 333: Only for neoglucobrassicin, which was a minor, dominant indolic glucosinolate in this crop, a 55% reduction was seen by increasing temperature from 4 to 18 °C (significant differences). Thus, it can be assumed no interfering effect by temperature for all the other biochemicals investigated in our study.

Line 332: Thus, we it can be assumed that there was no interfering effect by temperature for all the other biochemicals investigated ...

Line 353: To find the effective elicitors concerning phytochemicals increase in bBrassica leaves, we tested the following statistical hypotheses waswere tested for specific compounds...

Line 371: Thus, we could it can be concluded that the time delay between subsequent trial repetitions had no influence on results.

Line 457: The total content of ITCs in cabbage leaves in our study was about 3.15 µmol g-1d.w. (Figure 5B). 

Line 508: Our The results conclude that PAR treatment of leaves had no effect on antioxidant ....

Line 516: Our The investigation of the stimulating effect ...

Line 523: ....but our the present study increase, after a quite comparable UV exposure, was much higher ....

Line 526: The difference can also be explained by differences in radiation spectrum, as our the tested UV lamps (UVB-313 EL) lamps contained not only UV-B but also some UV-A, while the broccoli study used only UV-B (Figure 1, Table 1). 

Line 560-61: Since our the results indicate that sonication had no effect on the measured constituents, we it can be assumed that sonication at the given doses.... 

Line 565: Compared to the lettuce study, we used the the present study was performed with a higher intensity .......

line 572: ....so it could be assumed that thicker leaves need a more substantial exposure, higher than the levels we used, to respond to the treatment. 

Line 654: However, since we did not filter the small amounts of unwanted radiation were not filtered from lamps to have precise treatments of only PAR or UV, we such results cannot with certainty be concluded such results from our study. Our The results revealed hypotheses H1 to be rejected.

Line 686: We thank The authors gratefully acknowledge Prof. Knut Asbjørn Solhaug at.....

Comment 2.       Line 90-136. It is worth emphasizing the scientific novelty of the manuscript. This part is rewritten with the aim to make the novelty more clear:

Nutraceutical improvement of cabbage rest fractions have not been done previously by short-time ultrasound treatment (2-8 sec) alone or in combination with irradiation (10-14h) by UV and/or photosynthetic active radiation (PAR). The present study is an approach reducing losses in the food chain by adding value to side streams (crop waste fractions). Ultrasound was selected as an appropriate method based on its ability to enrich nutraceuticals in lettuce leaves within a very short elicitation period (1-3 sec) [21]. PAR and UV were included as well-known phytochemical elicitors in plants. Still, there is a lack of studies dealing with the utilisation of waste stream of pale inner leaves of stored cabbage (which has been little directly exposed to UV during the growth period). The biochemicals focused in the present study, phenols, GLS, isothiocyanates, and indoles, are among the most studied Brassica phytonutrients which have the potential to be modified by the selected elicitors.

New text:

The present study is an approach to reduce losses in the food chain by adding value to side streams (crop waste fractions), a topic of high interest for the food industry. Cabbage leaf rest fractions has not previously been studied with respect to phytochemical improvements by physical elicitor factors like ultrasound treatment (2-8 sec) in combination with UV irradiation (10-14h) or photosynthetic active radiation (PAR). Ultrasound was selected as an appropriate method based on its ability to enrich nutraceuticals in lettuce leaves within a very short elicitation period (1-3 sec) [21]. PAR and UV were included as well-known phytochemical elicitors for plants. Still, there is a lack of studies on enhancement of phytochemical concentration to waste streams of pale inner leaves of stored cabbage (with restricted radiation exposure during cultivation). The compounds being analyzed in the present study, phenols, GLS, isothiocyanates, and indoles, are among the most studied Brassica phytonutrients which have the potential to be modified by the selected elicitors.

Comment 3.       Are the processes of improving the quality of agro-waste described in the manuscript economically justified in the context of application in the food industry?

line 136-140: new part added to answer the comment: For the economic justification in the food industry, it is important to find methods with shortest possible time required for induction of the biochemical response, and to find methods which are possible to implement with industrial scale assembling bands or rotating drum systems where moving leaf parts are exposed continuously over time.

line 725: A new part is also added to the conclusion: To address the economic justification of such elicitor treatments in the food industry, further studies should investigate methods for repeated exposure of the rest raw materials over the required time-period for elicitation, for implementation in large scale equipment innovations like assembling bands or rotating lattice drum systems.

Comment 4. Has the "conclusion" section been omitted?

We understood from the author’s guidelines that the conclusion should be included as a part of the discussion, but we may have misunderstood this point. We have now moved our concussion to a separate Chapter no 5.

  1. Conclusion

The combination of PAR at 497 kJ m-2 + UV at 59 kJ m-2 was found to be the most effective elicitor combination by significantly increasing the content of phenolics and hydroxycinnamate monosaccharides by a doubling to tenfold increase, as vell as enhancing antioxidant activity by 30% in brassica leaf rest fraction. Ultrasound at 15-60 kJ l-1 did not affect the phytochemicals. Based on the promising results on lettuce found in the literature, higher energy levels of ultrasound should be tested in follow-up studies before making conclusions on possible elicitor effects of ultrasound on phytochemicals in cabbage leaves. GLS were not affected by UV, PAR or ultrasound, except for total indoles, which tended to decrease by UV treatment (59 kJ m-2). The tested abiotic elicitors seem to be a less relevant tool for increasing GLS content in cabbage rest raw materials. Nevertheless, they do not seem to be reduced when such treatment is used to increase other phytochemicals. For the economically justification in the food industry, further studies should investigate methods for repeatedly exposure to moving leaf parts over the required time period for elicitation. This will gain knowledge to implement in large scale equipment innovations like assembling bands or rotating lattice drum systems with repeatedly exposure of portions of the leaf material at a time.

Round 2

Reviewer 2 Report

The authors responded to all comments from the reviewer and improved the manuscript. I recommend publishing the manuscript.